# Practical Recommendations for the Manipulation of Kinase Inhibitor Formulations to Age-Appropriate Dosage Forms

**DOI:** 10.3390/pharmaceutics14122834

**Published:** 2022-12-17

**Authors:** Emma C. Bernsen, Valery J. Hogenes, Bastiaan Nuijen, Lidwien M. Hanff, Alwin D. R. Huitema, Meta H. M. Diekstra

**Affiliations:** 1Princess Máxima Center for Pediatric Oncology, Department of Pharmacology, Heidelberglaan 25, 3584 CS Utrecht, The Netherlands; 2Department of Pharmacy & Pharmacology, The Netherlands Cancer Institute—Antoni van Leeuwenhoek, Plesmanlaan 121, 1066 CX Amsterdam, The Netherlands; 3Department of Clinical Pharmacy, University Medical Center Utrecht, Heidelberglaan 100, 3584 CX Utrecht, The Netherlands

**Keywords:** pediatric oncology, manipulation, formulation, kinase inhibitor, bioequivalence

## Abstract

Over 75 kinase inhibitors (KIs) have been approved for the treatment of various cancers. KIs are orally administrated but mostly lack pediatric age-appropriate dosage forms or instructions for dose manipulation. This is highly problematic for clinical practice in pediatric oncology, as flexible oral formulations are essential to individually set dosages and to adjust it to a child’s swallowability. Most KIs are poorly soluble, categorized in Biopharmaceutics Classification System (BCS) class II or IV, and improperly manipulating the KI formulation can alter pharmacokinetics and jeopardize KI drug safety and efficacy. Therefore, the goals of this review were to provide practical recommendations for manipulating the formulation of the 15 most frequently used KIs in pediatric oncology (i.e., bosutinib, cabozantinib, cobimetinib, crizotinib, dabrafenib, dasatinib, entrectinib, imatinib, larotrectinib, nilotinib, ponatinib, ruxolitinib, selumetinib, sunitinib and trametinib) based on available literature studies and fundamental drug characteristics and to establish a decision tool that supports decisions regarding formulation manipulation of solid oral dosages of KIs that have been or will be licensed (for adult and/or pediatric cancers) but are not included in this review.

## 1. Introduction

Manipulating solid oral dosage forms of kinase inhibitors (KIs) without proper instructions can lead to a higher risk of over- and underdosing; unsafe situations for healthcare professionals, parents and patients; and a higher risk of feeding tube occlusions in pediatric oncology [1]. Since the introduction of imatinib in 2001, over 75 KIs have been approved for the treatment of various adult cancers [2]. These drugs are increasingly used off-label in pediatric oncology and often lack age-appropriate oral dosage forms [3,4,5]. This comprises a major challenge in pediatric oncology, as flexible oral formulations are essential since dosages are individually set (mostly based on body surface or weight) and need to be adjusted to a child’s swallowability [6]. Furthermore, it has recently been shown that of 58 oral targeted anticancer drugs (most of which were KIs), 11% had instructions for dose manipulation in the drug label [3]. This is especially problematic, as most KIs have poor aqueous solubility and are categorized as Biopharmaceutics Classification System (BCS) class II or IV, which increases the risk of altering KI pharmacokinetics (PK) and bioavailability when manipulating the KI formulation [7,8]. When no oral liquid or instructions are available, bioequivalence studies that investigate formulation manipulation of KIs can be used to ensure comparable in vivo performance of two medicinal products containing the same active substance. In these studies, the area under the curve (AUC), maximum plasma concentration (Cmax) and time to maximum plasma concentration (Tmax) serve as surrogate endpoints for drug safety and efficacy and are used to calculate the relative bioavailability. Bioequivalence is, in most cases, assumed when the AUC and Cmax of one medicinal product (e.g., KIs’ manipulated dosage form) are between 80% and 125% of the reference medicinal product (e.g., KIs’ solid oral dosage form). This is determined in bioequivalence or bioavailability studies that meet the requirements of the study design described in the European Medicines Agency (EMA) or U.S. Food and Drug Administration (FDA) bioequivalence guidelines [9,10].

However, most dose manipulations of KIs are not investigated in official bioequivalence studies but rather discussed in pediatric PK studies. These studies typically calculate PK parameters for pediatric patients that were treated with a solid oral dosage form or with the manipulated dosage form of the KI. This, however, is valuable information when no bioequivalence study is performed. In addition, stability studies, case reports, and the physicochemical properties and pharmacological characteristics of KIs can also support decisions regarding KI dose formulation manipulation to age-appropriate dosage forms.

In pediatric oncology, pharmacists and parents are frequently challenged to manipulate solid oral dosages forms of KIs to an age-appropriate form without proper instructions. This leads to unknown consequences on KI PK and bioavailability and, thus, drug efficacy and safety. Therefore, this review created an overview of literature and data relevant for dosage form manipulation and provided practical recommendations for the 15 most frequently used KIs in pediatric oncology (i.e., bosutinib, cabozantinib, cobimetinib, crizotinib, dabrafenib, dasatinib, entrectinib, imatinib, larotrectinib, nilotinib, ponatinib, ruxolitinib, selumetinib, sunitinib and trametinib), taking into account the possible risks of altering KI pharmacokinetics [11]. Secondly, we established a decision tool that supports manipulating solid oral dosages forms of other KIs that have or will be licensed (for adult and/or pediatric cancers) but are not included in this review.

## 2. Methods

### 2.1. Literature Analysis

In this review, evidence that supported practical recommendations for manipulating the formulation of 15 KIs were collected. The literature was searched for studies that described or investigated the manipulation of KI formulations in either bioequivalence, pediatric or adult PK studies and/or pediatric or adult case reports. A literature search was performed in PubMed, Cochrane and Embase (filtered by the period from 2000 to December 2021) including keywords and synonyms for “pediatrics”, “oncology”, “pharmacokinetics”, “bosutinib”, “cabozantinib”, “cobimetinib”, “crizotinib”, “dabrafenib”, “dasatinib”, “entrectinib”, “imatinib”, “larotrectinib”, “nilotinib”, “ponatinib”, “ruxolitinib”, “selumetinib”, “sunitinib”, “trametinib”, “formulation manipulation” and “enteral tube administration” (Appendix A). In addition, KI stability studies were collected. After the searches were conducted, duplicates were removed and the remaining articles were screened by title, abstract and full text based on our inclusion and exclusion criteria (Appendix B). Subsequently, cross-referencing was performed to include studies that were not found with the advanced literature search. Finally, EMA drug assessment reports, EMA summary of product characteristics (SmPC) and FDA drug labels were reviewed for available oral dosage forms, excipients, stability, physicochemical and solubility information.

### 2.2. Providing Practical Recommendations

The practical recommendations for manipulating the formulation of 15 KIs were based on the collected information from the literature analysis. We divided the practical recommendations into two categories: (A) manipulation of the solid oral dosage form to a compounded crushed tablet or opened capsule and (B) manipulation of the solid oral dosage form to an oral liquid.

For each KI per category, a level of evidence (LoE) was assigned (inspired by the Oxford Centre for Evidence-Based Medicine system) (Figure 1) [12]. The LoE represents the strength of evidence and the degree of uncertainty of altering PK parameters of KIs when preparing and administrating the manipulated dosage form. The recommendations were based on data that was found per KI, preferably from pediatric studies, but if these were not available, from studies performed with an adult population. The developed recommendations that were rated LoE 2–4 followed the applied dosage form manipulations performed in the studies found in the literature. LoE 1 was given if an alternative oral liquid formulation or an extemporaneous oral preparation instruction was available in the drug label, or when bioequivalence was demonstrated between a manipulated and solid oral dosage form of a KI in a study that met the requirements of the European Medicine Agency (EMA) and/or U.S. Food and Drug Agency (FDA) bioequivalence investigation guidelines [9,10]. LoE 2 was based on pediatric PK studies that showed similar PK parameters (i.e., within 80–125% of the solid oral form) of a manipulated formulation of a KI but did not perform a bioequivalence study that met the requirements of the EMA/FDA guidelines. LoE 3 was given to recommendations that were based on adult PK studies that showed similar PK parameters between two formulations of a KI (i.e., solid and manipulated formulation) but also did not met the requirements of the EMA/FDA guidelines. LoE 4 was given to recommendations that were based on information from case reports describing KI formulation manipulation (with or without PK data) in pediatric patients and/or adults. LoE 5 was given to recommendations solely based on theoretical considerations such as excipients, stability and physicochemical characteristics (i.e., BCS class, log P and solubility range) of the KI. If standardized excipients were used in the solid oral dosage form, the formulation was not modified and the product was chemically and physically stable (which can be found in the EMA assessment reports), the recommendation stated that it is possible to crush tablets, or to open capsules and sprinkle the content over spoon of a vehicle [7]. For recommendations that described the manipulation to an oral liquid, BCS class and aqueous pH solubility range were used to determine the dissolution content. For KIs with a wide pH range solubility, a neutral solvent (i.e., water) was chosen. For KIs with a low pH solubility (i.e., weak base), a low pH solvent was included in the recommendations. LoE 5 was also given to recommendations that were based on PK or bioequivalence studies confirming similar PK parameters between an unlicensed oral liquid formulation that was provided by the pharmaceutical company versus solid oral formulation in pediatric patients and/or adults [11].

## 3. Results

We included 15 EMA SmPCs, 15 EMA assessment reports and 1 FDA drug label. One pediatric PK study showed similar PK parameters between an solid oral dosage form and a manipulated formulation of a KI (sunitinib); one was a stability study (sunitinib); two were pediatric case reports (dabrafenib, trametinib and bosutinib); one was an adult case report (crizotinib); two were pediatric case reports including PK parameters but no manipulated oral dosage form (imatinib and ponatinib); one was an adult bioequivalence study between an unlicensed oral liquid formulation provided by the pharmaceutical company and the solid oral dosage form (trametinib); and twenty-four were pediatric phase I/II studies (including the earlier mentioned sunitinib PK study) [13,14,15,16,17,18,19,20,21,22,23,24,25,26,27,28,29,30,31,32,33,34,35,36,37,38,39,40,41,42,43,44,45,46,47,48,49,50,51,52]. Based on these findings, practical recommendations for manipulating the formulations to either a crushed/opened (solid) oral form or to an oral liquid were formulated and are shown in Table 1. An overview of pharmacological and physicochemical properties of the 15 KIs is presented in Table 2. In Appendix C, a summary of KI excipients can be found.

### 3.1. Phase I/II Study Results

Twenty-four phase I/II pediatric PK studies were found in the literature (Appendix D) [39,46,53,54,55,56,57,58,59,60,61,62,63,64,65,66,67,68,69,70,71,72,73,74,75]. The PK phase I/II studies with two formulations (either manipulated dosage form or licensed oral liquid), with the exception of larotrectinib and sunitinib, did not distinguish between the two formulations when calculating PK parameters and, thus, have not been able to reveal similar PK parameters and/or variability between the two oral dosage forms. Therefore, the information from these phase I/II studies was not further used in this review. Of the 15 KIs, we did not find phase I/II pediatric PK studies of cobimetinib, ponatinib, bosutinib entrectinib and trametinib.

### 3.2. LoE 1 Recommendation (Larotrectinib, Dasatinib, Imatinib, Ruxolitnib)

For four out of 15 KIs investigated, a licensed oral liquid (i.e., larotrectinib and dasatinib) or instructions for extemporaneous oral preparation in the EMA SmPC/FDA drug label (i.e., imatinib and ruxolitinib) was available [31,35,37,38]. As expected, no official bioequivalence studies were found.

### 3.3. LoE 2 Recommendation (Sunitinib)

Sunitinib was the only KI for which the practical recommendation was assigned LoE 2. One pediatric PK study that investigated AUC, Cmax, Tmax, half-life and toxicity of the registered oral dosage form, and the manipulated formulation of sunitinib was found in [39]. In this study, 12 pediatric oncology patients with refractory solid tumors were treated with sunitinib capsules that were opened and sprinkled onto applesauce or yoghurt. The median Tmax appeared earlier (4 h for the sprinkled content versus 7 h for capsules as a whole), but the Cmax, AUC and half-life of the manipulated dosage form was between 80 and 125% of the whole capsules [39]. In addition, a study by Sistla et al. (2004) showed that sunitinib is stable in a dissolved state in apple juice for at least two hours (within required specification of 95–105%) [40]. However, sunitinib is light-sensitive, and within two hours, the E isomer of sunitinib was formed as degradation product up to a level of 1.6%. On the basis of the stability and PK results, our recommendations to manipulate sunitinib capsules to either a sprinkled form or to an oral liquid were assigned an LoE of 2.

### 3.4. LoE 3 Recommendation

No recommendation was assigned an LoE 3.

### 3.5. LoE 4 Recommendation (Dabrafenib, Bosutinib, Crizotinib, Trametinib)

The practical recommendations of dabrafenib, trametinib, bosutinib and crizotinib to manipulate the solid dosage forms to either crushed tablets/opened capsules or to an oral liquid were based on case reports and assigned LoE 4 [41,43,44].

#### 3.5.1. Dabrafenib and Trametinib

The LoE of manipulating dabrafenib and trametinib formulations to an oral liquid was rated 4 [43]. We found one case report describing treatment with dabrafenib and trametinib in a 17-month-old patient who was diagnosed with high-grade glioneural tumor (with a BRAF V600E mutation). Because this patient had a gastrostomy tube, trametinib tablets and dabrafenib capsules were opened and dissolved in 5 mL water and administrated via the tube, resulting in a treatment response for at least six months [43]. This case report did not include PK parameters, but the results indicated that both dabrafenib and trametinib were adequately absorbed and can be dissolved and administrated without difficulties through a gastrostomy tube.

#### 3.5.2. Bosutinib

One case report described a four-year-old boy diagnosed with chronic myeloid leukemia (CML) and who was treated with crushed bosutinib tablets with a small amount of chocolate pasta after dinner [44]. The authors calculated PK parameters (Cmax, Cmin and AUC) at steady state for two different dosages (180 mg/day and 200 mg/day). Although these parameters were not similar to adults, a cytogenic (but no molecular) treatment response was seen in this patient.

#### 3.5.3. Crizotinib

One case report that manipulated crizotinib capsules to an oral liquid suspension was found in the literature [41]. This case report included a 68-year-old woman who was treated with crizotinib for a lung adenocarcinoma. Crizotinib was dissolved in 50 °C water and administrated via a nasogastric (and later percutaneous endoscopic gastrostomy (PEG)) tube. Subsequently, a therapeutic trough plasma concentration and an effective treatment response was seen in this patient. It is not clear if the authors dissolved the complete capsule (with shell) or opened the capsules, but we assume, because of the water temperature, that the whole capsule was dissolved and this was taken up in our recommendation.

### 3.6. LoE 5 Recommendations (Cobimetinib, Ruxolitinib, Cabozantinib, Dabrafenib, Imatinib, Ponatinib, Bosutinib, Crizotinib, Entrectinib, Nilotinib, Selumetinib, Trametinib)

No information was found for 12 recommendations (either changing the solid form to crushed/opened and/or an oral liquid) of cobimetinib, ruxolitinib, cabozantinib, dabrafenib, imatinib, ponatinib, bosutinib, crizotinib, entrectinib, nilotinib, selumetinib and trametinib. These practical recommendations were based on excipients used, stability and physicochemical characteristics (Table 2 and Appendix D) [13,16,17,23,25,29,30,32,34,47,49,52,76,77,78].

### 3.7. BCS Class I: Cobimetinib and Ruxolitinib

#### 3.7.1. Cobimetinib

As no literature studies or drug label information was available, the given recommendation is exclusively based on the information on the excipients used and physicochemical characteristics of cobimetinib. The formulation of cobimetinib tablets is not modified/enabled and does not include non-standardized excipients. In addition, cobimetinib is highly soluble over the gastrointestinal (GI) tract pH range. Therefore, a low risk of precipitation thus altering PK and bioavailability is expected when manipulating the dosage form to administrate it as crushed tablets or an oral liquid. This is included in the cobimetinib recommendations.

#### 3.7.2. Ruxolitinib

There is no information (in the drug label or in literature) available whether ruxolitinib tablets can be crushed [31,38]. The tablets contain standard excipients, the formulation is not modified/enabled and ruxolitinib is highly soluble in the gastrointestinal (GI) tract pH range. Therefore, if palatability is a problem and the oral liquid cannot be administrated, it is justified to either crush the tablets, or open the capsules and sprinkle the content over applesauce or chocolate pasta.

### 3.8. BCS Class II and IV

The uncertainty of the given recommendations on altering PK is higher for BCS class II and IV KIs (i.e., cabozantinib, dabrafenib, imatinib, ponatinib, bosutinib, crizotinib, entrectinib nilotinib, selumetinib and trametinib), thus these recommendations need to be considered with caution. Furthermore, the SmPC and EMA assessment reports of dabrafenib, entrectinib and selumetinib include warnings that should also be taken into account before staring a dose formulation manipulation. Dabrafenib is chemically instable, and no absorption of selumetinib in a suspension was observed, which led to an enabled formulation with a non-standardized excipient (i.e., solubilizing agent vitamin E polyethylene glycol succinate) to improve the solubility and absorption of selumetinib. In addition, entrectinib solubility is very pH-sensitive and includes a non-standardized excipient (i.e., acidulant) in the formulation to minimize effects of changing pH of the GI tract on absorption of entrectinib [16,30,47]. These warnings are included in the practical recommendations and were assigned an LoE 5.

#### 3.8.1. Trametinib

We found one bioequivalence between an unlicensed pediatric oral liquid formulation (provided by the pharmaceutical company) and tablets of trametinib in 16 adults with solid tumors [42]. However, we could not use this information to formulate a dose manipulation recommendation or to recommend using the pediatric oral liquid as it is not licensed.

#### 3.8.2. Ponatinib and Imatinib

One PK case report of ponatinib and imatinib has been found in the literature [45,46]. Although these case reports calculated PK parameters for the solid oral dosage form, it was not useful for this review as there was no information about PK parameters or information on a manipulated form of imatinib and ponatinib. The PK case report of imatinib (from 2009) included four children who were diagnosed with Philadelphia chromosome-positive (Ph+) leukemias. In this case, report, PK parameters (AUC and Cmax) of imatinib mesylate and metabolite N-desmethyl-imatinib (CGP 74588) were calculated [45]. The case report of ponatinib included one three-year-old patient who was diagnosed with Ph+ acute lymphatic leukemia (ALL) and was treated with ponatinib [46]. In this case report, plasma trough concentrations were calculated. Interestingly, the trough level of ponatinib varied in the patient during each treatment phase despite the same daily dose, indicating high intra-individual variability of ponatinib. Probably due to this variability and the ponatinib-induced toxicity seen in this patient, it was challenging to find the right ponatinib dose. Given the fact that ponatinib is a BCS class IV KI and is not soluble pH > 2, this could have contributed to this variability. How ponatinib was administrated is not mentioned in the case report.

## 4. Relevant Additional Recommendations

### 4.1. Therapeutic Drug Monitoring

A reliable option to account for possible altered PK and bioavailability when using a manipulated dosage form of KIs is therapeutic drug monitoring (TDM). We therefore recommend performing TDM after dose manipulation to confirm that the KI is being absorbed, by following the TDM targets proposed by Janssen et al. (2020) [79].

### 4.2. Administration Via Enteral Feeding Tube

We included practical recommendations to manipulate the solid dosage form to an oral liquid in view of the nasogastric feeding tubes that are commonly used in pediatric oncology. However, when considering administrating KIs via a feeding tube, the site, size, tube material and possible enteral food interactions need to be taken into account [81]. For example, the feeding tube site can influence drug bioavailability due to loss of solubility and different absorption windows. Most KIs are weak acids, and their dissolution step is dependent on the gastric acidity before it can be absorbed in the duodenum. It could therefore be a disadvantage to administrate KIs at a jejunal site due to the higher risk of fast precipitation and lower (or no) absorption of the KI in the jejunum [81]. In addition, the risk of drug/excipient adherence to the tube or tube occlusion and discomfort for the patient needs to be taken into account before administrating, although little is known about these risks for KIs. To mitigate this unknown risk, it is recommended to rinse the tube with water or with the vehicle (e.g., apple juice) that the KI is dissolved in before and after KI administration. Instructions for flushing and rinsing feeding tube before and after administration can be found in Table 1 or in Williams (2008) [81].

## 5. The Decision Tool

Figure 2 presents the decision tool that can be used by pharmacists as a guideline for manipulating KI formulations not included in this review. The colors show the potential and unknown risk of altering stability, solubility, pharmacokinetics (Cmax, AUC, Tmax) and bioavailability of KIs when manipulating the solid dosage form. These risks (i.e., no, low, medium and high risk) are integrated in the general recommendations. Green represents a low risk, followed by yellow, orange and red representing potential high and unknown risks. Low risk was given to recommendations that follow the information in the drug leaflet or advise to use an licensed oral liquid. A low to median risk was given to recommendations that need to be based on literature studies that assessed and showed similar PK parameters between the manipulated and solid oral dosages forms of KIs. However, assessing the risk of the manipulated oral form of KIs becomes more complicated when no information in literature or the drug label is available. Here, the pharmacist has to formulate recommendations that are based on the excipient, stability and physicochemical properties of a KI. We consider it high risk to manipulate KI formulations if the product (active pharmaceutical ingredient (API) and excipients) is chemically and/or physically unstable or if the KI formulation is modified or enabled. If this is not the case, pharmacists can use the BCS class and pH-dependent solubility (both can be found in the KIs’ EMA assessment report) to choose a suitable content and/or dissolution vehicle. KIs in BCS classes I and III have a high solubility and a resp. high and low permeability. These BCS class drugs will rapidly dissolve in a wide pH range; thus, manipulating the formulation to an oral liquid will most likely not lead to relevant differences in PK and bioavailability as these drugs already have a high bioavailability or, in the case of BCS class III KIs, the bioavailability will be limited due to low permeability (i.e., the possibility of a drug to be transported over a membrane by transporter proteins), not solubility [7,11,95]. In contrast, BCS class II and IV KIs will show more variability in PK and bioavailability due to their low solubility, which is the limiting factor for these drugs to be absorbed [8]. Because of these differences in BCS classes, the general recommendation for BCS class I and III KIs were considered median risk of altering bioavailability, safety and stability as these drugs will easily dissolve and the impact on bioavailability will probably be lower than for BCS class II and IV [95,96]. The general recommendations for BCS classes II and IV were considered high risk, as these KIs have a low solubility leading to a higher variability in PK and bioavailability and possible risk of precipitation before being able to be absorbed [8,95]. If no information on solubility is available, the chemical properties (i.e., weak base or acid) can be used to decide the solvent vehicle.

## 6. Discussion

In this review, we provided practical recommendations for manipulating the formulation of 15 KIs used in pediatric oncology. In addition, we developed a decision tool that can guide pharmacists whenever they are challenged to manipulate the solid oral dosage form of KIs that are not discussed in this review. Out of the 15 KIs, 4 KIs (i.e., larotrectinib, dasatinib, imatinib and ruxolitinib) are available as oral liquid or include instructions in the drug label. Sunitinib was the only KI for which a pediatric PK study of a manipulated dosage form was performed [39]. For ten KIs (cobimetinib, cabozantinib, dabrafenib, ponatinib, bosutinib, crizotinib, entrectinib, nilotinib, selumetinib and trametinib), no sufficient evidence was found for the given recommendations and these were based on case reports, and physicochemical and pharmacological properties of the KI. As expected, these results are in accordance with recent studies that have showed the lack of age-appropriate oral liquids or instructions for dose manipulations in drug labels of KIs [1,3,7,97].

A remarkable finding was that the pediatric phase I/II PK studies of ruxolitinib, dasatinib, imatinib, nilotinib and sunitinib included manipulated oral dosage forms (Appendix D) that, according to the EMA guideline on requirements to the chemical and pharmaceutical quality documentation concerning investigational medicinal products in clinical trials, section 4.2.1.P underwent compatibility and stability testing [98]. Highly regrettable, this information is not taken up into the drug labels or otherwise publicly available; the EMA drug labels of sunitinib and imatinib do not mention that the capsule can be opened and sprinkled over yoghurt or applesauce, and the ones of ruxolitnib, dasatinib and nilotinib do not include any information about the manipulated dosage forms that were used in these studies. We also found discrepancies between drug assessment reports and EMA/FDA drug labels and widely scattered information about the manipulation of KI formulations. For example, the EMA assessment report of trametinib states that trametinib insoluble, but in the literature, it was reported that trametinib can be dissolved in water. Similarly, the relevant information of the PK and stability study of sunitinib has not been included in the drug label but is found in the literature and the FDA ruxolitinib drug label included instructions to develop an oral liquid of ruxolitinib, but this is not mentioned in the EMA ruxolitinib drug label [24,38,39,40]. What was even more striking was that the phase I/II PK studies of crizotinib, dabrafenib and trametinib included an oral liquid formulation that was provided by the pharmaceutical company but are not authorized on the drug market [99,100].

The recommendations provided in Table 1 were limited by the small amount of data that was available and most were rated LoE 4 and 5. Without evidence, formulation manipulation of KIs will lead to unknown alterations of the drug properties (such as drug solubility and permeability) and, therefore, PK and bioavailability. Additionally, the manipulation of crushing tablets, opening capsules and dissolving the drug could induce unexpected excipient–drug interactions that can further affect KI PK and palatability and acceptability in children, which are important aspects of KI efficacy, safety and treatment adherence [101]. To account for the limited amount of data available and these unknown effects on PK and bioavailability, TDM is highly recommended to further guide KI dosing and to visualize the possible effects of the manipulated dosage forms on KI PK and bioavailability [79].

The LoE categories were mainly driven by evidence on bioequivalence or PK studies investigating manipulated and solid oral dosage forms, as information on stability was mostly lacking. The developed recommendations were, therefore, intended for immediate use, thereby limiting risks on changes in physicochemical and microbiological stability. This, however, complicated assigning the sunitinib recommendation as this was based on combined information from one PK study and one stability study as this both contributed to knowledge about safety of manipulating sunitinib to an oral liquid.

A limitation of the decision tool (Figure 2) is that the BCS class and solubility range that was used as a risk indicator for altering PK and bioavailability could be misleading. For example, we argued that low pH solubility and BCS classes II–IV were high-risk drugs when altering the formulation, but we possibly overestimated the risk of altering PK and bioavailability of low pH solubility KIs as these (in both dosage forms) will precipitate in the small intestines (absorption window) where the pH ranges from 6 to 8. A possible overlooked higher risk in the decision tool than mentioned is that the absorption of BCS class I KIs could increase due to a longer absorption window as it is already in its soluble form, which consequently could lead to more toxicity [102].

Alternative drug-delivery methods are essential in pediatric oncology, but most KIs do not have an age-appropriate dosage form [3]. This hampers the use of KIs in pediatric oncology, while KIs have a promising new role in several pediatric cancer treatments, including relapse or refractory tumors [4,103,104]. As PK and bioequivalence studies in pediatric oncology cohorts are challenging and limited in availability, forthcoming research should focus on dissolution tests to evaluate the dissolution and stability of manipulated KI dosage forms in different gastro-intestinal conditions to account for age that, in combination with TDM of KIs, could be used as an alternative for bioequivalence studies (and act as a BCS-based biowaiver) [95,105]. Additionally, as official bioequivalence studies need large sample sizes and are time-consuming, an additional accelerated route to safer use of manipulated KI dosage forms could be to design small PK studies (with, e.g., ten pediatric oncology patients or adults) to confirm efficacy and safety between two (i.e., solid and manipulated) KI formulations [106]. However, this is secondary to the responsibility of pharmaceutical companies to develop age-appropriate dosage forms of KIs.

## 7. Conclusions

This review reflects how scarce information on formulation manipulation for clinical care of pediatric oncology is. It is crucial that pharmaceutical companies develop age-appropriate dosage forms of KIs and that the unlicensed oral liquids of crizotinib, dabrafenib and trametinib that have been used in previous and current clinical trials, become authorized on the drug market. In addition, information on dose manipulation, stability and compatibility of manipulated KI formulations used in the early pediatric clinical trials should become publicly available. This will highly contribute to safer and effective KI treatment in pediatric oncology patients. Until that is available, this review supports decision making in clinical practice on formulation manipulation of KIs for pediatric use. TDM after manipulation of the KIs’ solid dosage form is highly recommended to ensure safe and effective treatment.

## Figures and Tables

**Figure 1 pharmaceutics-14-02834-f001:**
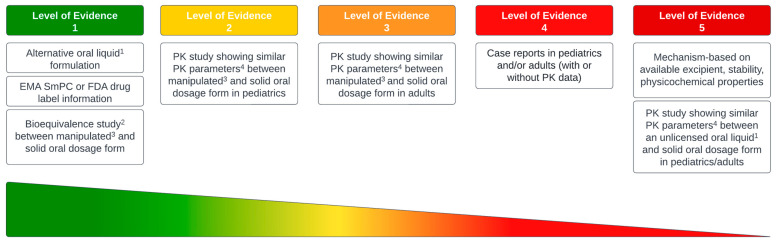
Category of practical recommendations according to their level of evidence. ^1^ oral liquid developed by the pharmaceutical company; ^2^ Study that follows the requirements of the EMA and/or FDA bioequivalence investigation guidelines; ^3^ manipulated (i.e. capsule opened, tablets crushed and/or dissolved); ^4^ similar PK parameters = Area Under the Curve (AUC) and maximum plasma concentration (Cmax) of the manipulated oral dosage form are within 80–125% of the PK parameters of the solid oral dosage form; EMA = European Medicines Agency; FDA = U.S. Food and Drug Administration; SmPC = summary of product characteristics; PK = pharmacokinetics.

**Figure 2 pharmaceutics-14-02834-f002:**
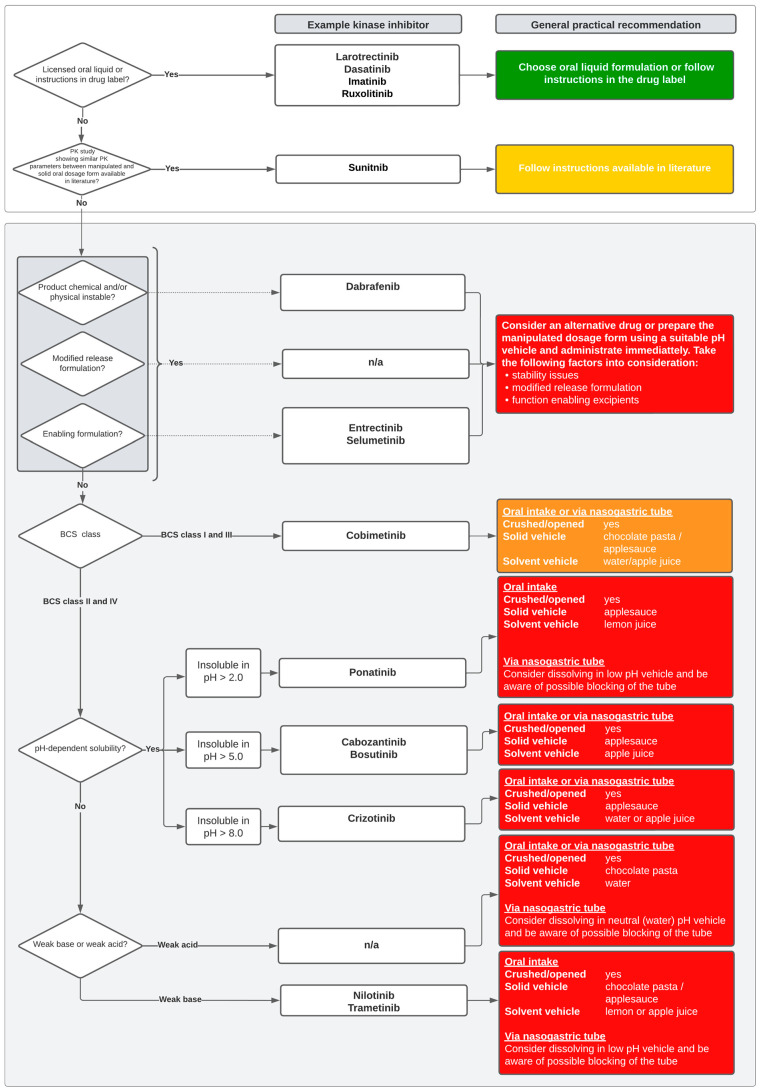
Decision tree for manipulating solid oral dosage forms of KIs. The colors represent the unknown risk of altering safety, stability, pharmacokinetics (AUC, Cmax) and bioavailability when administrating the manipulated dosage form (green = low risk, red = high risk); Crushed/opened = can tablet be crushed capsule be opened; Solid vehicle = possible vehicle to add to crushed tablets or content of capsules; Solvent vehicle = possible vehicle to dissolve tablets/capsule in; BCS = Biopharmaceutics Classification System; AUC = Area Under the Curve; Cmax = maximum plasma concentration; Tmax = time to maximum plasma concentration.

**Table 1 pharmaceutics-14-02834-t001:** Practical recommendations for the manipulation of a KI formulation.

	Crushed/Opened (Solid) Oral Administration (A)	LoE A	Oral Liquid (i.a. Nasogastric Tube Administration 1) (B)	LoE B	Refs
**BCS class I**					
**Cobimetinib**	Crush the tablet and administrate with a small amount of apple sauce or chocolate pasta	5	Dissolve the tablet in water and carefully stir/shake until the suspension is formed. Administrate the suspension immediately	5	[13,32,76]
**Larotrectinib**	Use available oral solution	1	Use available oral solution	1	[37]
**Ruxolitinib**	Crush the tablet and administrate with a small amount of apple sauce	5	Dissolve tablet in 40 mL of water and carefully stir/shake for 10 min. Administrate the suspension immediately	1	[24,38]
**BCS class II**					
**Cabozantinib**	Open capsule or crush the tablet and sprinkle content over a spoon of apple sauce	5	Dissolve tablet or content of capsule in apple juice and carefully stir/shake. Administrate the suspension immediately	5	[17,34,77]
	**Cave**: food interaction (higher AUC, Cmax and risk of toxicity)				
**Dabrafenib**	Open the capsule and sprinkle content over a spoon of applesauce. Administrate the prepared suspension immediately	5	Dissolve capsule content in 5 mL water or apple juice and carefully stir/shake. Administrate the suspension immediately	4	[30,43,52,78]
	**Cave**: food interaction and chemical instable				
**Dasatinib**	Use available oral suspension	1	Use available oral suspension	1	[35]
	**Cave**: antacid interaction, no bioequivalence between tablets and oral suspension				
**Imatinib**	Open capsule or crush the tablet and sprinkle content over a spoon of applesauce	5	Dissolve 100 mg imatinib (capsule content or tablet) with 10 mL water. Administrate the suspension immediately	1	[31,45]
	**Cave**: do not mix imatinib with orange juice, milk or cola				
**Ponatinib**	Crush the tablet and administrate with a small amount of applesauce	5	Dissolve tablet in lemon juice and carefully stir/shake. Administrate the suspension immediately	5	[15,19,46]
	**Cave**: only soluble in pH ≤ 2.0				
**BCS class IV**					
**Bosutinib**	Crush tablet in a small amount of chocolate pasta	4	Dissolve tablet in apple juice and carefully stir/shake. Administrate the suspension immediately	5	[33,44,50]
	**Cave**: antacid interaction				
**Crizotinib**	Open the capsule and sprinkle the content over a small amount of applce sauce	5	Dissolve capsule (with shell) in warm water (50 °C) and carefully stir/shake. Administrate the suspension immediately	4	[28,41]
**Entrectinib**	Open the capsule and sprinkle the content over small amount of applesauce	5	Consider dissolving the capsule content in a low pH vehicle (e.g., lemon juice)	5	[29,47]
	**Cave**: not soluble and has non-standardized excipient (acidulant) in the formulation. Be aware of possible blocking of the feeding tube				
**Nilotinib**	Open the capsule and sprinkle content over a spoon of apple sauce or chocolate pasta	5	Consider dissolving the capsule content in a low pH vehicle (e.g., lemon juice) due to insolubility of nilotinib	5	[23,25]
	**Cave**: food interaction and insolubility of nilotinib. Be aware of possible blocking of feeding tube				
**Selumetinib**	Open the capsule and sprinkle the content over a small amount of applesauce	5	Consider dissolving the capsule content in a low pH vehicle (e.g., lemon juice)	5	[16,49]
	**Cave**: food interaction, low absorption in suspension formulation, use of non-standardized excipient (solubilizing agent) and insolubility of selumetinib. Be aware of possible blocking of feeding tube				
**Sunitinib**	Open the capsule and sprinkle the content over applesauce (or yoghurt)	2	Dissolve capsule content in apple juice and carefully stir/shake. Administrate the suspension immediately	2	[26,39,40]
	**Cave**: sunitinib is light sensitive and will form (impure) isomers within two hours after dissolving in a glass of apple juice				
**Trametinib**	Crush tablet and add content to a spoon of apple sauce	5	Dissolve tablet in 5 mL water or consider dissolving the tablet in a low pH vehicle (e.g., lemon or apple juice) and carefully stir. Administrate the suspension immediately	4	[21,27,42,43]
	**Cave**: food interaction and insolubility of trametinib. Be aware of possible blocking of feeding tube				
**General recommendations** Always perform therapeutic drug monitoring (TDM) after manipulating the solid dosage form (see TDM recommendations in study of Janssen et al., 2020 [79]). Always use gloves and a mouth cap before manipulating the solid dosage form of anticancer drugs [80].					

^1^ See Williams (2008) [81] for detailed instructions: Instruction flushing and rinsing feeding tube before and after administration; 0–1 year: 2–5 mL dissolution vehicle; 1–16 year: 5–10 mL dissolution vehicle; ≥16 years: 10–20 mL dissolution vehicle. Tablets or capsules can be dissolved in 15–30 mL dissolution vehicle unless otherwise mentioned. Always dissolve the content in an oral syringe that can be attached to the feeding tube. BCS = Biopharmaceutics Classification System; LoE = level of evidence; AUC = area under the curve; Cmax = maximum plasma concentration.

**Table 2 pharmaceutics-14-02834-t002:** Physicochemical properties and pharmacological characteristics of KIs.

	Off-Label Indication	Ligand	Log *p*	Solubility Range (pH)	pK_a_	Salt Form	Refs
**BCS Class I**							
Cobimetinib	Solid tumors	MEK	3.9	1.0–7.5	-	Hemifumarate	[13,76]
Larotrectinib	Solid tumors, Primary CNS tumors	TRK	1.7	1.0–8.0	-	Sulfate	[48,82]
Ruxolitinib	Relapsed or refractory solid tumors, leukemia or myeloproliferative neoplasms	JAK	2.1	1.0–8.0	0.91, 5.51, 13.89	Phosphate	[20,83]
**BCS Class II**							
Cabozantinib	HB, HCC	VEGF	5.4	1.0–4.0 (capsules) 1.0–3.0 (tablets)	-	Malate	[17,18,77]
Dabrafenib	LGG, HGG	B-RAF	4.8	1.0–4.0	−1.5	Mesylate (and micronized, pharmaceutical development)	[52,78]
Dasatinib	CML	BCR-ABL	3.6	-	-	Monohydrate	[36,84]
Imatinib	CML, ALL	BCR-ABL	3.5	Soluble in water	-	Mesilate	[14,85]
Ponatinib	CML, Ph+ ALL	BCR-ABL	4.1	1.0–2.0	2.77, 7.8	HCL	[15,86]
**BCS Class IV**							
Bosutinib	CML	BCR-ABL	5.4	1.0–5.0	-	Monohydrate	[50,87,88]
Crizotinib	ALCL	ALK	1.65	1.6–8.2	-	na	[22,89]
Entrectinib	Solid tumors	ROS 1 and NTRK	5.7	Not soluble	-	na	[47,90]
Nilotinib	Relapsed or refractory malignancies	BCR-ABL	4.9	-	2.1, 5.4	Monohydrate	[25,91]
Selumetinib	Relapsed or refractory tumors	MEK	3.6	Not soluble	-	Hydrogen sulfate	[16,92]
Sunitinib	Renal tumors	VEGF	5.2	1.0–5.0	8.95	Maleate	[51,93]
Trametinib	LGG, HGG	MEK	3.4	Not soluble	-	Dimethyl sulfoxide	[21,94]

ALCL = anaplastic large-cell lymphoma; ALL = acute lymphoblastic leukemia; CML = chronic myeloid leukemia; CNS = central nervous system; HB = hepatoblastoma; HCC = hepatocellular carcinoma; HGG = high-grade glioma; LGG = low-grade glioma, Ph + ALL = Philadelphia chromosome-positive acute lymphoblastic leukemia; MEK = mitogen-activated protein kinase; TRK = tropomyosin receptor kinase; JAK = janus kinase; VEGF = vascular endothelial growth factor; ALK = anaplastic lymphoma kinase; ROS = reactive oxygen species; NTRK = neurotrophic tyrosine receptor kinase.

## Data Availability

Not applicable.

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
