# Peer review of "Practical Recommendations for the Manipulation of Kinase Inhibitor Formulations to Age-Appropriate Dosage Forms"

_pharmaceutics, 2022, doi:10.3390/pharmaceutics14122834_

Round 1

Reviewer 1 Report

The manuscript provides practical recommendations for adapting the formulation of 15 kinase  inhibitors used in pediatric oncology  based on available literature studies and drug characteristics. In addition, a decision tool is presented to guide decisions regarding formulation manipulation of solid oral dosages of other KIs. This is an important contribution to clinical practice despite the limitations discussed in the paper.

In the discussion, it would be important to highlight that also the methods used to manipulate the solid dosage forms can impair PK and bioavailability. For instance, crushing the tablets can induce drug adsorption on excipients such as microcrystalline cellulose, the dissolution of capusle shells changes the solution composition putting shell excipients in close contact with drug.

Have you taken these possibilities into consideration? What about the risk involved?

Author Response

Dear reviewer 1,

Thank you for your useful comments and suggestions for our manuscript entitled " Practical recommendations for the manipulation of kinase inhibitor formulations to suitable pediatric dosage forms".

Please see below our response and changes in our manuscript.

“In the discussion, it would be important to highlight that also the methods used to manipulate the solid dosage forms can impair PK and bioavailability. For instance, crushing the tablets can induce drug adsorption on excipients such as microcrystalline cellulose, the dissolution of capusle shells changes the solution composition putting shell excipients in close contact with drug. Have you taken these possibilities into consideration? What about the risk involved?”

Thank you for this useful comment. We included this in the discussion section as we agree this is a valuable insight for the readers which was not yet included in the manuscript. Please see manuscript lines 376-386

For revisions based on reviewers’ comments and suggestions, please see lines 111-113 and 134-135.

For minor additional revisions:

Title                            3

Abstract                     17-18

Introduction               44-46, 50, 64-68, 72-73, 76

Methods                      135

Table 1                        165

Results                         283-286

Discussion                   350, 353-375, 387-389, 427-445

Conclusion                  448-457

Please let us know if further revisions are necessary for our manuscript. Thank you again and we are looking forward to hearing your final publication decision.  

Yours sincerely,

Authors Manuscript Pharmaceutics-2028170

Reviewer 2 Report

This is a comprehensive and informative review on pediatric dosage form for kinase inhibitors (KI). The authors performed a literature search and gave practical recommendations on the common KIs and decision logic for formulation manipulation.

Author Response

Dear reviewer 2,

Thank you for your rating of our manuscript entitled " Practical recommendations for the manipulation of kinase inhibitor formulations to suitable pediatric dosage forms".

For revisions based on reviewers’ comments and suggestions, please see lines 111-113, 134-135 and 376-386.

For minor additional revisions:

Title                            3

Abstract                     17-18

Introduction               44-46, 50, 64-68, 72-73, 76

Methods                      135

Table 1                        165

Results                        283-286

Discussion                   350, 353-375, 387-389, 427-445

Conclusion                  448-457

Please let us know if further revisions are necessary for our manuscript. Thank you and we are looking forward to hearing your final publication decision.  

Yours sincerely,

Authors Manuscript Pharmaceutics-2028170

Reviewer 3 Report

The present manuscript deals with interesting topics of preparation dosage forms with selected kinase inhibitors suitable for pediatric patients from dosage forms available for adults. The topic is important and research work adds some new information for preparation of dosage forms appropriate for application to pediatric patients.

General suggestion: The study is well organized, but it I suggest authors to add additional information in Table 1 and discussion how dose adjustment is performed during manipulation of dosage forms intended for adults. From the presented data it is not clear whether dose adjustment was applied in the citated PK studies.

Suggested corrections:

Page 4, line 131: replace “dissolvent” with “solvent”;

Author Response

Dear reviewer 3,

Thank you for your useful comments and suggestions for our manuscript entitled " Practical recommendations for the manipulation of kinase inhibitor formulations to suitable pediatric dosage forms".

Please see below our response and changes in our manuscript.

“General suggestion: The study is well organized, but it I suggest authors to add additional information in Table 1 and discussion how dose adjustment is performed during manipulation of dosage forms intended for adults. From the presented data it is not clear whether dose adjustment was applied in the citated PK studies.”

Thank you for this suggestion. The recommendations rated LoE 2-4 follow the applied dose adjustments found in the literature. As Table 1 is quite dense with information, we included this comment in the method section. Please see manuscript lines 111-113.

“Page 4, line 131: replace “dissolvent” with “solvent””

Thank you for this correction. We revised this in the manuscript, please see line 134-135

For revisions based on reviewers’ comments and suggestions, please see lines 376-386.

For minor additional revisions:

Title                            3

Abstract                     17-18

Introduction               44-46, 50, 64-68, 72-73, 76

Methods                      135

Table 1                        165

Results                        283-286

Discussion                   350, 353-375, 387-389, 427-445

Conclusion                  448-457

Please let us know if further revisions are necessary for our manuscript. Thank you again and we are looking forward to hearing your final publication decision.  

Yours sincerely,

Authors Manuscript Pharmaceutics-2028170

Round 2

Reviewer 3 Report

No additional comments.